# Mental Health Outcomes and Occupational Stress Among Malaysian Frontline Workers During the COVID-19 Pandemic

**DOI:** 10.3390/healthcare13202584

**Published:** 2025-10-14

**Authors:** Nadia Mohamad, Siti Sara Yaacob, Rohaida Ismail, Imanul Hassan Abdul Shukor, Mohd Zulfinainie Mohamad, Muhammad Farhan Mahmud, Mohd Faiz Ibrahim

**Affiliations:** 1Environmental Health Research Centre, Institute for Medical Research, National Institutes of Health, Shah Alam 40170, Selangor, Malaysia; nadia@moh.gov.my (N.M.); rohaidadr@moh.gov.my (R.I.); drfaizibrahim@moh.gov.my (M.F.I.); 2Department of Public Health Medicine, Faculty of Medicine, Universiti Teknologi MARA, Sungai Buloh 47000, Selangor, Malaysia; sitisara@uitm.edu.my; 3Non-Communicable Disease Control Sector, Selangor State Health Department, Ministry of Health Malaysia, Shah Alam 40100, Selangor, Malaysia; mohdzulfi@moh.gov.my (M.Z.M.); m.farhan@moh.gov.my (M.F.M.)

**Keywords:** workplace stress, health workforce, crisis response, psychosocial impact, occupational health

## Abstract

**Background:** The COVID-19 pandemic significantly affected the mental health of frontline healthcare workers. This study investigated the mental health and occupational stressors faced by frontline workers in Selangor during the pandemic. **Methods:** A cross-sectional study was conducted using secondary data from the Mental Health and Psychosocial Support Services team, collected from March to August 2020. A total of 4593 frontline workers participated in the Depression, Anxiety, and Stress Scale screening. **Results:** Mental health symptoms were common among frontline workers during the COVID-19 pandemic, with 14.8% reporting stress, 30.7% anxiety, and 20.4% depression. Female workers had significantly higher odds of all three conditions, with adjusted odds ratios (aOR) of 1.35 (95% CI: 1.10–1.66) for stress, 1.25 (95% CI: 1.07–1.47) for anxiety, and 1.23 (95% CI: 1.03–1.47) for depression. Workers aged 18–30 had higher odds of stress (aOR 1.88; 95% CI: 1.42–2.47), anxiety (aOR 1.74; 95% CI: 1.43–2.12), and depression (aOR 1.80; 95% CI: 1.43–2.27) compared with those over 40. Employment in hospitals was associated with increased odds of all three conditions, with aORs ranging from 1.71 to 2.05. Among 711 respondents who reported occupational stressors, lack of mental health support was the strongest predictor (aORs 4.91–5.20), followed by poor work rotation and conflict with supervisors. **Conclusions:** Women, younger staff, and hospital workers were more vulnerable to mental health symptoms during the pandemic. Organizational factors, especially limited support and poor work arrangements, played a major role. Targeted mental health services and improved working conditions are needed to support healthcare workers in future emergencies.

## 1. Introduction

Public health emergencies are events that can overwhelm the routine capabilities of communities to manage health consequences arising from their causative agents. Throughout the history of mankind, the world has witnessed numerous such events, dating back as early as 430 Before the Common Era (B.C.E.), when the Athenian Plague struck the region of Ethiopia. This unknown novel pathogen led to the deaths of one-fourth of the Athenian population, scattered over 400 years [1]. In the mid-14th century, the human species experienced the deadliest pandemic in history, known as the Black Death. It was a type of bubonic plague that resulted in an estimated 200 million casualties across the continents of Europe, Asia, and Africa [2]. The latest pandemic, coronavirus disease 2019 (COVID-19), originated from the city of Wuhan, China, causing an exponential increase in healthcare demands, resulting in straining the world’s healthcare system to its breaking point [3]. Planet Earth was then at a standstill, with strict movement control orders in many parts of the world, affecting human daily lives and activities.

As healthcare demands skyrocketed during pandemics, the immense burden on the healthcare system systematically exposed healthcare workers to occupational stress [4]. During the COVID-19 pandemic, there was an 80% increase in the occupational workload among the healthcare workers [5]. Moreover, a study in the United States revealed that 61% of healthcare workers feared the risks of exposure to the virus, 43% suffered from work overload, 49% experienced burnout, and more than one-third had to deal with anxiety or depression [6]. In addition, higher stress levels are found among frontline workers, such as nursing and medical assistants, inpatient and social workers. Similar findings have been observed in countries such as Singapore, China, Saudi Arabia, and Italy [7,8,9,10].

In the context of Malaysian healthcare workers, some articles have discussed mental health within the population [11,12]. However, limited studies addressed mental health stressors among frontline workers [13]. With the increasing likelihood of future outbreaks due to emerging or reemerging diseases, it is crucial to understand and address the mental health status of frontline workers. This is essential to maximize the mental preparedness of the health workforce, subsequently to minimize the effects of occupational stressors, and to prevent any untoward mental health crises during such public health emergencies [14,15]. Therefore, this study aimed to examine the mental health status of frontline workers and its associated factors in Malaysia during the recent COVID-19 pandemic. By identifying key stressors and their associated mental health outcomes, this research seeks to inform the Ministry of Health with regard to the strategies to support and protect the well-being of frontline workers during current and future public health emergencies.

## 2. Materials and Methods

### 2.1. Study Area and Population

Selangor is the most populous state in Malaysia, with a recorded population of approximately 7 million as of the 2020 census [16]. Strategically positioned around Kuala Lumpur, Selangor plays a central role in Malaysia’s economy and transportation infrastructure, including hosting the Kuala Lumpur International Airport (KLIA) and Port Klang, Malaysia’s busiest seaport and a major gateway for international trade. The state’s robust transportation network and high concentration of economic activity contribute to significant internal migration and high population mobility [17]. During the COVID-19 pandemic, Selangor emerged as one of the hardest-hit regions in Malaysia. It consistently reported the highest number of confirmed COVID-19 cases nationwide, a trend attributed to its high population density, economic activity, and traveler influx. Consequently, Selangor hosted the largest number of government-designated quarantine centers to manage incoming travelers and high-risk contacts during the containment phase [12]. Given this high disease burden and the operational pressure on local health infrastructure, healthcare workers (HCWs) in Selangor faced exceptional occupational challenges.

#### Mental Health and Psychosocial Support Services (MHPSS) Program

During the COVID-19 pandemic, mental health screening and psychological support was provided by the Mental Health and Psychosocial Support Services (MHPSS) team in Selangor. The program offered a mental health screening to the full eligible population of frontline personnel across public healthcare facilities. It broadly represents the workforce composition of frontline personnel in Selangor, encompassing various occupational groups with mostly female workers. It is not surprising, as the majority of the healthcare workforce in Malaysia, across all professions, is predominantly female [18]. The screening was universally applied to the large coverage and inclusion of staff across clinics, hospitals, and district health offices to enhance the contextual relevance of the frontline population. The frontline workers are ground-level employees who engage directly with the public or provide health services that are essential to the population. These workers refer to health and social care workers and volunteers from various organizations, with diverse educational and professional backgrounds.

Mental health outcomes were assessed using the Depression Anxiety and Stress Scale 21 (DASS-21), a validated tool for measuring depression, anxiety, and stress, with seven items per subscale [19]. Both the English and Malay versions of the DASS-21 demonstrate good internal consistency reliability among local populations, with Cronbach’s alpha values ranging from 0.74 to 0.96 [19,20,21,22]. Responses were rated on a scale from 0 (never) to 3 (most or all the time), with total scores categorized into none, mild, moderate, severe, and extremely severe.

Occupational stressors included (1) concerns about workplace safety, refer to workers’ concern about their protection at work including enough personal protective equipment or infection prevention and control measure provided; (2) unsatisfactory work environments, and lack of recognition refers to inadequate welfare facility at work and no recognition or reward given for their contribution; (3) inadequate manpower, refer to not enough staff with increasing workload; (4) poor work rotation, refers to poor working hours, or have to work overtime; (5) conflict with supervisors, refers to lack of communication with supervisor or disagreement with supervisor; (6) inconsistent leave policies, refers either difficulty to apply leaves or their leave were cancelled; and (7) lack of mental health support, refers to lack of support or counselling provided. Responses were rated as either “Yes” or “No”.

### 2.2. Data Collection

This study utilized secondary data obtained from mental health screening and psychological support activities conducted by the MHPSS team, which consist of data on (1) sociodemographic characteristics, (2) mental health outcomes, and (3) occupational stressors. A universal sampling approach was applied to the MHPSS database, whereby all the registered mental health screenings between March and August 2020 were included in the data collection. A total of 6795 registered data from the MPHSS database were collected for this study. After excluding 2192 data due to duplicates and incomplete information, the final sample comprised 4593 data. Among these, only 711 have complete data for both the DASS and occupational stressors study. These 711 data were all included in the analysis of occupational stressors and mental health outcomes (Figure 1).

Data were analyzed using IBM SPSS Statistics (Version 29.0) analytics software. Descriptive statistics for sociodemographic characteristics and mental health outcomes were presented as frequencies and percentages for categorical data and means and standard deviations (±SD) for continuous data. Statistical tests were two-sided, with *p* < 0.05 considered significant. Simple and multiple logistic regression analyses were conducted to explore the effects of various categorical variables (gender, age, ethnicity, healthcare facility, Ministry of Health (MOH) staff, job category, and involvement in outbreak management) on mental health outcomes (stress, anxiety, and depression) using the ENTER method. Independent variables were coded as 1 for reference, and multicollinearity was checked. Model fitness was assessed using the Hosmer–Lemeshow test and classification table.

## 3. Results

The results reveal a predominantly female workforce (75.1%) with an average age of 34 years. The majority are Malay (81.0%), working mainly in hospitals (59.1%). Nearly all are MOH staff (98.6%). Mid-level staff formed the largest group (51.4%). A substantial majority (70.9%) were directly involved in outbreak management. Table 1 describes the characteristics of the sample population in detail.

Statistical analysis showed no multicollinearity in all the modes with a variance inflation factor (VIF) < 5, indicating no strong correlation between the variables. The models were a good fit based on Hosmer and Lemeshow tests (*p*-value > 0.05), and the overall classified percentage is acceptable.

There were significant associations between ethnicity, type of healthcare facilities, MOH staff, and job category with mental health status from the crude analysis (Table 2). Malay workers had lower odds of stress and depression compared with non-Malay workers, while MOH staff had lower odds of stress, anxiety, and depression compared with non-MOH staff. However, when adjusted with other factors, both factors had no significant difference in mental health status.

After adjusting for other variables, gender differences were notable, with female workers showing higher odds of stress (AOR = 1.4, 95%CI = 1.0–1.7), anxiety (AOR = 1.3, 95%CI = 1.1–1.5), and depression (AOR = 1.2, 95%CI = 1.0–1.5). Younger workers (18–30 years) experienced higher levels of stress (AOR = 1.9, 95%CI = 1.4–2.5), anxiety (AOR = 1.7, 95%CI = 1.4–2.1), and depression (AOR = 1.8, 95%CI = 1.4–2.3). Respondents working in hospitals and state/district health offices experienced higher stress, anxiety, and depression than those in clinics. The job category was another important factor, with low-level support staff experiencing the least mental health issues. Frontline workers who were involved directly in outbreak management were found to have 22% more odds for stress than frontline workers who were not directly involved with outbreak management.

Table 3 shows occupational stressors and mental health among the frontline workers. The occupational stressors in this study were divided into three groups: working environment, human resources, and organizational support. In multiple logistic regression, poor work rotation and lack of mental health support were associated with mental health. Poor work rotation had higher odds of stress (AOR = 3.1, 95%CI = 1.5–6.5), anxiety (AOR = 2.4, 95%CI = 1.2–4.9), and depression (AOR = 3.2, 95%CI = 1.6–6.5), while lack of mental health support also showed more odds of stress (AOR = 4.9, 95%CI = 2.7–9.1), anxiety (AOR = 5.2, 95%CI = 2.7–9.9), and depression (AOR = 4.4, 95%CI = 2.4–8.0). In contrast, stressors that had no significant association with all three mental health statuses were an unsatisfactory work environment, lack of recognition, and inconsistent leave policies (*p*-value > 0.05).

## 4. Discussion

This study underscores the significant mental health challenges faced by frontline workers in various healthcare settings during the COVID-19 pandemic in Selangor, Malaysia. The findings highlighted that female frontliners were more susceptible to stress, anxiety, and depression. The psychological burden borne by women in healthcare during outbreaks is well documented, and the COVID-19 pandemic has accentuated these gender disparities [23,24,25]. Our results align with findings from a study in Malawi, which reported comparable levels of anxiety among female healthcare workers but a higher prevalence of depression relative to our current study [15]. The heightened impact on women is likely attributable to increased professional workloads compounded by additional caregiving responsibilities at home, particularly in the context of restricted access to schools and childcare services during the pandemic [22,26]. A comprehensive meta-analysis has demonstrated that nations with higher levels of gender inequality had more severe mental health issues among female healthcare workers, encompassing 22 countries across South America, Europe, Asia, and Africa [27]. Although Malaysia was not specifically categorized among the highest-inequality nations in that analysis, the findings provide useful context for interpreting the current study’s results, especially considering the added domestic and family responsibilities that many women managed during the pandemic.

Another notable finding was the impact of workplace stressors across all age groups, with the highest odds ratio observed in the younger age group of 18–30 years. These findings were consistent with studies conducted in Spain and the United States, which also found an association between young age and mental health issues such as anxiety, depression, post-traumatic stress disorder (PTSD), and sleep pattern disturbances [28,29]. This suggested that younger healthcare workers may be less equipped to handle the intense pressures brought about by the pandemic, possibly due to a combination of less professional experience and fewer coping mechanisms [30,31]. Older individuals and experienced healthcare workers may have better psychological strengths acquired from life-challenging experiences, equipping them with skills to deal with the crisis of COVID-19 [32,33]. The lack of human interaction due to movement restrictions further exacerbates stress among younger workers, leading to loneliness and mental health issues [34,35]. These findings emphasize the need for age-specific mental health support to help younger healthcare workers cope with the challenges of the pandemic.

When comparing different levels of workplace environments, frontline workers working as state and district health officers exhibited higher odds of experiencing mental health issues. This disparity is likely attributable to the higher workload in the public health sector, which includes responsibilities such as contact tracing, screening, and surveillance—tasks that have been especially demanding during the COVID-19 emergency [36]. They are often at the forefront of managing public health crises, dealing with an overwhelming influx of cases and the logistical challenges of implementing and maintaining public health measures [37]. Furthermore, the responsibility of screening large populations adds another layer of pressure, as these health officers must ensure accurate and timely testing to control the spread of the virus [38]. Health inspectors working in district health offices have been observed to have the highest prevalence of work-related burnout compared with other healthcare workers in Malaysia [39]. In addition to their regular duties, these health inspectors were heavily involved in the COVID-19 screening, contact tracing, decontamination processes, and response in handling of deceased bodies. Surveillance duties also demand constant vigilance and rapid response to emerging hotspots, further amplifying the mental strain on these workers.

During the COVID-19 crisis, the healthcare system was overwhelmed, and a small number of non-MOH staff, many of them serving as volunteers, stepped in to assist the health sector. Many were assigned to quarantine centers and did not necessarily have a medical background. Although non-MOH staff comprised a small proportion of our sample (1.4%), their data were retained to provide insight into the mental health experiences of this distinct group. They showed higher odds of stress, anxiety, and depression, though estimates should be interpreted with caution due to the limited sample size. Similar findings have been reported elsewhere [40,41]. In Spain, many volunteers reported feeling unprepared due to insufficient supervision, expressed fears of getting infected and infecting their relatives or friends, leading to higher levels of stress, anxiety, and depression [40]. Moreover, a cross-sectional study conducted in the United States and China uncovered intriguing results [41]. Volunteers experienced higher levels of mental distress, including depression, anxiety, and somatization. However, they also reported greater levels of happiness compared with those who did not volunteer. The findings of the current research highlighted several critical stressors affecting frontline workers during the national respiratory outbreak, which aligned with results from previous studies. Lack of concerns about workplace safety was a cause for anxiety and depression, which, in contrast with previous studies, indicates that frontline workers often feel vulnerable to infection [15,33]. Lack of concern may be attributed to lack of awareness, training, and perception of infection precautions, subsequently affecting mental health outcomes [42,43]. Poor work rotation had emerged as a significant stressor, mirroring findings from studies in the United States and other countries, which showed that inadequate scheduling and lack of rest breaks contributed to higher psychological stress [44,45,46]. This highlighted the importance of improved work management practices to ensure frontline workers are not overburdened. Furthermore, this current study revealed that the lack of mental health support was significantly associated with higher levels of stress, anxiety, and depression. This is in line with research from Canada and Slovenia, which highlighted the crucial role of mental health resources and support systems for healthcare workers during public health emergencies [47,48]. Addressing these key stressors is essential to mitigating adverse mental health effects.

## 5. Recommendation

To address the significant mental health challenges faced by the frontline workers during public health emergencies, such as disease outbreaks and pandemics, we recommend that the Ministry of Health strengthen the existing mental health support systems within healthcare facilities. This includes making mental health screenings more frequent and proactive, strengthening support networks, and ensuring easier access to professional counselling services. Additionally, workplace safety protocols must be enhanced, and work rotation schedules should be optimized to prevent poor mental health outcomes.

## 6. Limitations

This study was conducted in a single state, which may not fully represent other regions in Malaysia. The cross-sectional design captures a snapshot in time and does not allow for assessment of changes in mental health status over the course of the pandemic. Longitudinal studies are needed to better understand how mental health outcomes evolve over time and the lasting impact of public health emergencies. In addition, most participants were female, reflecting the gender distribution of the healthcare workforce in the setting studied. Nonetheless, this gender imbalance should be considered when interpreting the results, particularly in relation to gender-based comparisons. Finally, the small proportion of participants who responded to the question on perceived stressors may limit the ability to draw broader conclusions about the key factors affecting mental health among frontline workers.

## 7. Conclusions

This study highlighted the need for an adequate mental health support team to ensure mental health wellness during outbreaks, especially among young female hospital workers. While COVID-19 is no longer a significant threat to public health, this study added new knowledge on the mental health aspects of frontline workers during outbreaks and pandemics. Key stressors included concerns about workplace safety, poor work rotation, and lack of mental health support, all significantly associated with higher levels of stress, anxiety, and depression. These findings underscore the need for targeted mental health support, improved workplace safety protocols, and better work management practices to mitigate the mental health impact on healthcare workers during public health emergencies.

## Figures and Tables

**Figure 1 healthcare-13-02584-f001:**
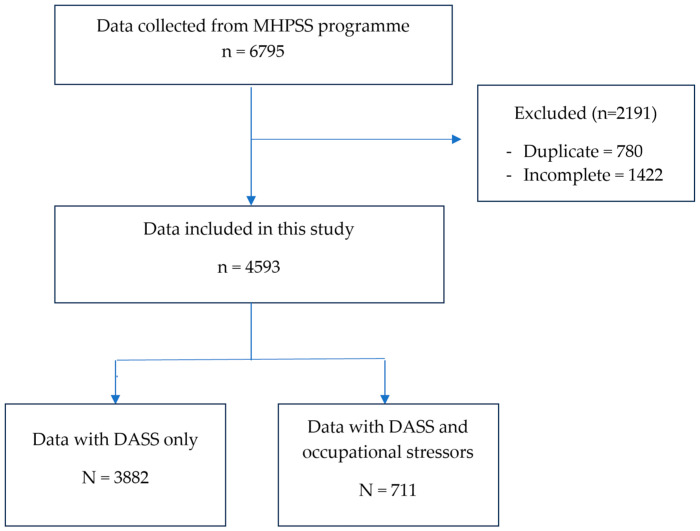
Flowchart of the data of frontline workers that was included in the study.

**Table 1 healthcare-13-02584-t001:** Characteristics of Frontliners at MOH facilities during the COVID-19 pandemic.

Variables	Mean (SD)	n	%
**Gender**			
Male		1145	24.9
Female		3448	75.1
**Age**	34.27 ± 7.29		
30 and below		1661	36.2
31–40		2137	46.5
>40		795	17.3
**Ethnicity**			
Malay		3722	81.0
Chinese		268	5.8
Indian		414	9.0
Others		189	4.1
**Healthcare facilities**			
Hospitals		2716	59.1
Clinics		1580	34.4
Health Offices		297	6.5
**Ministry of Health Staff**			
Yes		4530	98.6
No		63	1.4
**Job category**			
Support staff		811	17.7
Mid-level staff		2375	51.4
Professional staff		1366	29.7
Others		41	0.9
**Direct involvement in outbreak** **management**			
Yes		3255	70.9
No		1338	29.1

**Table 2 healthcare-13-02584-t002:** Factors associated with mental health issues/physiological conditions in MOH facilities (N = 4593).

Variable		Stress			Anxiety			Depression	
	Yes	Crude	Adjusted	Yes	Crude	Adjusted	Yes	Crude	Adjusted
	n (%)	OR (95% CI)	OR (95% CI)	n (%)	OR (95% CI)	OR (95% CI)	n (%)	OR (95% CI)	OR (95% CI)
**Gender**									
Male	152 (13.3)	1	1	331 (28.9)	1	1	234 (20.4)	1	1
Female	528 (15.3)	0.85 (0.70–1.03)	1.35 (1.10–1.66) *	1078 (31.3)	1.12 (1.07–1.47) *	1.25 (1.07–1.47) *	750 (21.8)	1.09 (0.92–1.28)	1.23 (1.03–1.47) *
**Age**									
18–30	315 (19.0)	2.25 (1.72–2.94) **	1.88 (1.42–2.47) **	613 (36.9)	1.85 (1.53–2.24) **	1.74 (1.43–2.12) **	453 (27.3)	2.11 (1.69–2.63) **	1.80 (1.43–2.27) **
31–40	290 (13.6)	1.51 (1.15–1.97) **	1.44 (1.10–1.89) *	605 (28.3)	1.25 (1.04–1.51) **	1.28 (1.06–1.55) *	413 (19.3)	1.35 (1.08–1.68) *	1.30 (1.04–1.63) *
>40	75 (9.4)	1	1	191 (24.0)	1	1	120 (15.1)	1	1
**Ethnicity**									
Malay	515 (13.8)	1	1	1134 (30.5)	1	1	765 (20.6)	1	1
Non-Malay	165 (18.9)	1.46 (1.20–1.77) **	1.20 (0.97–1.47)	275 (31.6)	1.05 (0.90–1.24)	1.01 (0.85–1.20)	221 (25.4)	1.31 (1.11–1.56) **	1.08 (0.90–1.30)
**Healthcare facilities**									
Clinics	157 (9.9)	1	1	362 (22.9)	1	1	223 (14.1)	1	1
Hospital	471 (17.3)	1.90 (1.57–2.31) **	1.90 (1.56–2.31) **	938 (34.5)	1.78 (1.54–2.05) **	1.71 (1.48–1.98) **	687 (25.3)	2.06 (1.75–2.43) **	2.05 (1.73–2.43) **
State/district health office	52 (17.5)	1.92 (1.37–2.71) **	2.00 (1.39–2.87) **	109 (36.7)	1.95 (1.50–2.54) **	2.01 (1.52–2.65) **	76 (25.6)	2.09(1.56–2.82) **	2.18 (1.60–2.98) **
**MOH staff**									
No	25 (39.7)	3.89 (2.33–6.49) **	1.62 (0.71–3.69)	29 (46.0)	1.95 (1.18–3.21) *	0.998(0.472–2.11)	27 (42.9)	2.79 (1.69–4.62) **	1.45 (0.68–3.12)
Yes	655 (14.5)	1	1	1380 (30.5)	1	1	959 (21.2)	1	1
**Job category**									
Support staff	71 (8.8)	1	1	219 (27.0)	1	1	121 (14.9)	1	1
Mid-level staff	330 (13.9)	1.68 (1.28–2.20) **	1.47 (1.11–1.93) *	747 (31.5)	1.24 (1.04–1.48) *	1.10 (0.92–1.32)	476 (20.0)	1.43 (1.15–1.78) **	1.26 (1.003–1.57) *
Professional staff	258 (18.9)	2.43 (1.84–3.21) **	2.09 (1.56–2.79) **	420 (30.7)	1.20 (0.99–1.46)	1.09 (0.89–1.34)	368 (26.9)	2.10 (1.68–2.64) **	1.90 (1.50–2.42) **
Others	21 (51.2)	10.94 (5.67–21.15) **	5.92 (2.21–15.88) **	23 (56.1)	3.45 (1.83–6.52) **	2.67 (1.06–1.25) *	21 (51.2)	5.99 (3.15–11.38) **	3.34 (1.31–8.51) *
**Outbreak management**									
No	195 (14.6)	1	1	418 (31.2)	1	1	286 (21.4)	1	1
Yes	485 (14.9)	0.97 (0.81–1.17)	1.22 (1.01–1.48) *	991 (30.4)	0.96 (0.84–1.11)	1.08 (0.94–1.25)	700 (21.5)	1.01 (0.86–1.18)	1.17 (0.99–1.37)

Note: * *p*-value < 0.05; ** *p*-value < 0.005; CI = confidence interval; OR = odds ratio. Hosmer–Lemeshow not significant (*p* > 0.05). Overall, correctly classified percentage > 69.5%.

**Table 3 healthcare-13-02584-t003:** Occupational stressors and mental health status (N = 711).

Stressor	Variable		Stress			Anxiety			Depression	
		Yes	Crude	Adjusted	Yes	Crude	Adjusted	Yes	Crude	Adjusted
		n (%)	OR (95% CI)	OR (95% CI)	n (%)	OR (95% CI)	OR (95% CI)	n (%)	OR (95% CI)	OR (95% CI)
Working environment	Concern on safety at workplace									
	No	117 (19.0)	2.21 (1.08–4.52)	1.67 (0.81–3.46)	230 (37.3)	2.20 (1.31–3.70) **	1.77 (1.04–3.01) *	161 (26.1)	2.41 (1.28–4.54) *	1.93 (1.01–3.66) *
	Yes	9 (9.6)	1	1	20 (21.3)	1	1	12 (12.8)	1	1
	Unsatisfactory work environment and lack of recognition									
	No	120 (17.9)	1.27 (0.52–3.09)	1.06 (0.42–2.65)	235 (35.1)	0.94 (0.49–1.80)	0.78 (0.40–1.53)	165 (24.6)	1.35 (0.61–2.98)	1.12 (0.50–2.52)
	Yes	6 (14.6)	1	1	15 (36.6)	1	1	8 (19.5)	1	1
Human resource	Inadequate manpower and increased workload									
	No	118 (17.8)	1	1	228 (34.4)	1	1	160 (24.2)	1	1
	Yes	8 (16.3)	0.90 (0.41–1.97)	1.11 (0.5–2.48)	22 (44.9)	1.55 (0.86–2.79)	1.89 (1.04–3.44) *	13 (26.5)	1.13 (0.58–2.19)	1.37 (0.70–2.70)
	Poor work rotation									
	No	114 (16.9)	1	1	232 (34.3)	1	1	157 (23.2)	1	1
	Yes	12 (34.3)	2.57 (1.24–5.32) *	3.10 (1.47–6.53) **	18 (51.4)	2.03 (1.03–4.01) *	2.43 (1.21–4.86) *	16 (45.7)	2.78 (1.40–5.54) **	3.22 (1.59–6.50) **
	Conflict with supervisor									
	No	111 (16.9)	1	1	233 (34.0)	1	1	156 (23.8)	1	1
	Yes	15 (27.3)	1.84 (0.98–3.45)	2.03 (1.06–3.89) *	27 (49.1)	1.87 (1.08–3.25) *	2.06 (1.17–3.65) *	17 (30.9)	1.43 (0.79–2.61)	1.57 (0.85–2.92)
Organization support	Inconsistent leaves policies									
	No	118 (17.7)	1	1	233 (35.0)	1	1	159 (23.9)	1	1
	Yes	8 (17.8)	1.00 (0.46–2.21)	1.23 (0.55–2.77)	17 (37.8)	1.13 (0.61–2.10)	1.37 (0.72–2.60)	14 (31.1)	1.44 (0.75–2.77)	1.71 (0.87–3.36)
	Lack of mental health support									
	No	103 (15.6)	1	1	215 (32.5)	1	1	146 (22.1)	1	1
	Yes	23 (46.0)	4.62 (2.55–8.36) **	4.91 (2.66–9.08) **	35 (70.0)	4.84 (2.59–9.06) **	5.20 (2.74–9.85) **	27 (54.0)	4.14 (2.31–7.44) **	4.39 (2.41–8.01) **

Note: * *p*-value < 0.05; ** *p*-value < 0.005; CI = confidence interval; OR = odds ratio. Hosmer–Lemeshow not significant (*p* > 0.05), overall, correctly classified percentage > 67.5%.

## Data Availability

The datasets generated and/or analyzed during the current study are not publicly available due to privacy and third-party restrictions.

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
