# Peer review of "Mental Health Outcomes and Occupational Stress Among Malaysian Frontline Workers During the COVID-19 Pandemic"

_healthcare, 2025, doi:10.3390/healthcare13202584_

Round 1
Reviewer 1 Report (Previous Reviewer 3)
Comments and Suggestions for Authors
No further comment. I kindly suggest to revise the result section in abstract to include a description of your studied population following the ORs for the risk factors you mentioned. Currently, it is a qualitative description.
Author Response
Comments:
No further comment. I kindly suggest to revise the result section in abstract to include a description of your studied population following the ORs for the risk factors you mentioned. Currently, it is a qualitative description.
Response:
We have revised the Results section of the abstract to include a description of the studied population following the reported odds ratios. (lines 25-32)
Reviewer 2 Report (Previous Reviewer 2)
Comments and Suggestions for Authors
Authors have accounted for prior reviewer comments and made the necessary amendments.
These are some additional comments:
a. Authors should describe any ethical considerations and approaches they took into account. Additionally, was an ethical approval process initiated?
b. A brief description of the research design should be noted.
c. Methods - describe the procedures for the study. If this was described in a previous publication, provide a reference/s for this
Author Response
Authors have accounted for prior reviewer comments and made the necessary amendments.
These are some additional comments:
a. Authors should describe any ethical considerations and approaches they took into account. Additionally, was an ethical approval process initiated?
Response: We did obtain ethical approval from the Medical Research Ethics Committee (MREC), Ministry of Health Malaysia, as stated in line 315-321.
b. A brief description of the research design should be noted.
Response: We have refined our materials and methods section, describing the research design in detail in lines 111-141.
c. Methods - describe the procedures for the study. If this was described in a previous publication, provide a reference/s for this
Response: The procedures for this study have been added in detail and described in lines 97-155.
Reviewer 3 Report (New Reviewer)
Comments and Suggestions for Authors
The revised manuscript investigates the mental health and occupational stressors faced by frontline workers during the COVID-19 pandemic. The revisions have strengthened the manuscript overall; however, several areas still require clarification and improvement. Below are detailed, section-specific comments:
Abstract:
- Please include key statistical values to provide readers with a quick overview of the study's findings and significance.
Introduction:
- Pg. 2, lines 66-67: The authors state that there are limited studies addressing mental health stressors within frontline workers, yet without providing proper citations to support their claim.
- Pg. 2, line 75: The phrase “stakeholders of the strategies” is vague. Please clarify who the stakeholders are in this context. Government? NGO? State?
Materials and methods:
- Were data used for this study collected FROM or BY MHPSS? The first paragraph states that data were collected by the MHPSS team, while the second states that the data were collected from them. Please clarify the actual data collection and the responsible party for the collection.
- It would be helpful to include background information about the country and/or the state of Selangor, where the study was conducted, to contextualize the findings.
- Pg. 3, lines 78-81: The opening sentence is confusing; please rephrase.
- Pg. 3, lines 108-110: Please clarify why only 711 out of 4,593 respondents had complete data for both the DASS and occupational stressors assessments.
- Pg. 4, line 119: The Ministry of Health is mentioned here for the first time. The authors should keep the full name followed by the acronym to ensure clarity for readers unfamiliar with the acronym.
Results:
- Pg. 5, Table 1: There is a significant gender imbalance in the sample, with 75.1% of respondents identifying as female. This limitation should be acknowledged and discussed in the Limitations section.
- Additional context is needed to understand why female workers are disproportionately represented in healthcare occupations. Is this reflective of the broader national healthcare workforce or specific to the state of Selangor, only?
- Pg. 6, line 143: Given that 98.6% of the sample identifies as MOH staff, the comparison between MOH vs non-MOH workers should be reconsidered or justified.
- Pg. 6, line 156: Please add the sample size used in the occupational stressors analysis.
Discussion:
- Line 187: Is Malaysia one of the nations with higher levels of gender inequality? Needs clarification.
- Pg. 2, lines 221-225: The authors report that volunteers experienced higher odds of mental health issues, but it is unclear in comparison to which reference group. These findings are not presented in the Results section or tables. Please include the relevant data and specify the number/proportion of these in the study sample.
Recommendation:
- Pg. 2, line 252: The term "stakeholders" remains unclear. Please specify who is being referenced here.
Limitations:
- As noted under the Results section, the overrepresentation of female participants should be acknowledged as a limitation.
Conclusion:
- The section heading should be singular, "Conclusion" instead of plural.
- Pg. 2, line 266: As "a crucial reference" to what? It is vague and needs clarification.
Author Response
The revised manuscript investigates the mental health and occupational stressors faced by frontline workers during the COVID-19 pandemic. The revisions have strengthened the manuscript overall; however, several areas still require clarification and improvement. Below are detailed, section-specific comments:
Abstract:
● Please include key statistical values to provide readers with a quick overview of the study's findings and significance.
Response: We have revised the Results section of the abstract to report the odds ratios. (lines 25-32)
Introduction:
● Pg. 2, lines 66-67: The authors state that there are limited studies addressing mental health stressors within frontline workers, yet without providing proper citations to support their claim.
Response: Thank you for highlighting. We have added relevant citation into the text. (line 66)
● Pg. 2, line 75: The phrase “stakeholders of the strategies” is vague. Please clarify who the stakeholders are in this context. Government? NGO? State?
Response: Thank you for highlighting. We have replaced ‘stakeholders’ with ‘the Ministry of Health’. (line 74)
Materials and methods:
● Were data used for this study collected FROM or BY MHPSS? The first paragraph states that data were collected by the MHPSS team, while the second states that the data were collected from them. Please clarify the actual data collection and the responsible party for the collection.
Response: This study is secondary data collection. The data were collected from the MHPSS team. We have refined it in the text and added a subheading ‘Data collection’ to improve the clarity in lines 132-141.
● It would be helpful to include background information about the country and/or the state of Selangor, where the study was conducted, to contextualize the findings.
Response: We have added a subheading titled “Study Area and Population” that provides background information on Selangor to better contextualize the study findings. (lines 79-92)
● Pg. 3, lines 78-81: The opening sentence is confusing; please rephrase.
The sentence has been revised for clarity (line 79)
● Pg. 3, lines 108-110: Please clarify why only 711 out of 4,593 respondents had complete data for both the DASS and occupational stressors assessments.
Response: It is stated in the study under the data collection section (page 4, lines 137-141).
●Pg. 4, line 119: The Ministry of Health is mentioned here for the first time. The authors should keep the full name followed by the acronym to ensure clarity for readers unfamiliar with the acronym.
Response: Thank you for the comment. We have added the Ministry of Health in the first mention (line 151)
Results:
● Pg. 5, Table 1: There is a significant gender imbalance in the sample, with 75.1% of respondents identifying as female. This limitation should be acknowledged and discussed in the Limitations section.
Response: We appreciate this observation. The gender imbalance has now been discussed in the revised Limitations section (lines 291-294) with a note on how this may affect the interpretation of gender-based findings
● Additional context is needed to understand why female workers are disproportionately represented in healthcare occupations. Is this reflective of the broader national healthcare workforce or specific to the state of Selangor, only?
Response: We added in the manuscript (page 3, Line 100-103)
● Pg. 6, line 143: Given that 98.6% of the sample identifies as MOH staff, the comparison between MOH vs non-MOH workers should be reconsidered or justified.
Response: We have retained the MOH vs non-MOH comparison to provide insight into the mental health status of non-MOH personnel, many of whom were volunteers without formal medical training but were involved in outbreak response. Although non-MOH staff comprised only 1.4% of our sample, we believe their inclusion is relevant given their distinct role during the pandemic. We have added a justification in the Discussion section (page 9, lines 247–252) and have advised cautious interpretation of the findings due to the limited sample size.
● Pg. 6, line 156: Please add the sample size used in the occupational stressors analysis.
Response: We added the sample sizes for occupational stressor analysis in the Materials and Methods section (page 4, line 140-141)
Discussion:
● Line 187: Is Malaysia one of the nations with higher levels of gender inequality? Needs clarification.
Response: We have clarified this point in the revised manuscript (page 9, lines 212–216).
●Pg. 2, lines 221-225: The authors report that volunteers experienced higher odds of mental health issues, but it is unclear in comparison to which reference group. These findings are not presented in the Results section or tables. Please include the relevant data and specify the number/proportion of these in the study sample.
Response: We apologise for the confusion. The group previously referred to as “volunteers” corresponds to non-MOH staff in our dataset. This terminology has now been clarified in the revised manuscript. (lines 247-249)
Recommendation:
●Pg. 2, line 252: The term "stakeholders" remains unclear. Please specify who is being referenced here.
Response: Thank you for highlighting. We have replaced ‘stakeholders’ with ‘the Ministry of Health’. (line 74)
Limitations:
●As noted under the Results section, the overrepresentation of female participants should be acknowledged as a limitation.
Response: We have addressed the overrepresentation of female participants in the revised Limitations section, lines 291-294.
Conclusion:
● The section heading should be singular, "Conclusion" instead of plural.
Response: Thank you for highlighting the section heading; it has been changed as recommended. (line 298)
● Pg. 2, line 266: As "a crucial reference" to what? It is vague and needs clarification.
Response: Thank you for highlighting. We have further clarified this issue with a clearer statement (lines 301-303)
Reviewer 4 Report (New Reviewer)
Comments and Suggestions for Authors
This article focuses on the mental health status of frontline workers in Malaysia during the COVID-19 pandemic, a topic with practical significance and social value. However, there are certain limitations in the research design, such as the use of a cross-sectional study, which makes it difficult to reveal the dynamic changes in mental health issues. It is recommended that the author consider incorporating a longitudinal tracking design in future studies to more accurately assess the long-term impact of public health emergencies on the mental health of frontline workers.
Regarding sample selection, there are limitations in sample representativeness. Although the sample size is large (4,593 individuals), the article does not sufficiently explain whether the sampling process was random and representative, especially whether the distribution of different occupational categories, genders, and age groups was balanced. It is recommended that the author discuss the representativeness of the sample to enhance the credibility of the study's results.
The DASS-21 scale used in the study is an effective tool for assessing mental health status, but no reliability and validity tests for the Chinese or Malay versions of the scale are provided. It is recommended that the author include information on the applicability of this version in the local population, such as Cronbach’s α coefficient and other statistical indicators, to ensure the scientific validity of the measurement tool. Furthermore, research variables such as "conflicts with supervisors" and "poor shift arrangements" lack clear definitions and operational indicators. It is suggested that the author further refine the variable definitions and explain how they were measured.
In terms of data analysis, the article uses multiple logistic regression analysis but does not report the results of multicollinearity tests among the variables, nor does it explain whether the regression model assumptions were met. It is recommended to supplement the model diagnostics, such as VIF values, residual analysis, etc., to improve the reliability of statistical inferences. Additionally, some conclusions do not closely correspond to the data analysis results. It is suggested to strengthen the logical connection between data analysis and conclusions to avoid excessive inferences.
Overall, this article provides valuable insights into the mental health issues faced by frontline workers during public health emergencies. However, to improve the quality of the research, it is recommended that the author make systematic revisions to the research design, variable operationalization, statistical methods, and conclusion derivation. Future studies could also consider integrating qualitative interview methods to further explore the specific stress experiences and coping strategies of frontline workers, thereby providing a basis for developing more targeted mental health intervention measures.
Author Response
This article focuses on the mental health status of frontline workers in Malaysia during the COVID-19 pandemic, a topic with practical significance and social value. However, there are certain limitations in the research design, such as the use of a cross-sectional study, which makes it difficult to reveal the dynamic changes in mental health issues. It is recommended that the author consider incorporating a longitudinal tracking design in future studies to more accurately assess the long-term impact of public health emergencies on the mental health of frontline workers.
Response: Thank you for highlighting this important limitation. We agree that a cross-sectional design limits the ability to assess changes in mental health over time. This has now been acknowledged in the revised Limitations section (lines 288-291), and we have recommended that future studies adopt a longitudinal design to capture the long-term mental health effects of public health emergencies.
Regarding sample selection, there are limitations in sample representativeness. Although the sample size is large (4,593 individuals), the article does not sufficiently explain whether the sampling process was random and representative, especially whether the distribution of different occupational categories, genders, and age groups was balanced. It is recommended that the author discuss the representativeness of the sample to enhance the credibility of the study's results.
response:
We have added clarification in the Materials and Methods section (page 4, lines 132-141) that this study employed a universal sampling approach, including all frontline workers who participated in the MHPSS screening in Selangor. While the sampling was not random, it covered a broad group of healthcare personnel from various facilities and job categories.
The DASS-21 scale used in the study is an effective tool for assessing mental health status, but no reliability and validity tests for the Chinese or Malay versions of the scale are provided. It is recommended that the author include information on the applicability of this version in the local population, such as Cronbach’s α coefficient and other statistical indicators, to ensure the scientific validity of the measurement tool.
response:
We added the reliability in the Materials and Methods section (page 3, line 113-115)
Furthermore, research variables such as "conflicts with supervisors" and "poor shift arrangements" lack clear definitions and operational indicators. It is suggested that the author further refine the variable definitions and explain how they were measured.
response:
We added the definition in the Materials and Methods section (pages 3-4, line 119-129)
In terms of data analysis, the article uses multiple logistic regression analysis but does not report the results of multicollinearity tests among the variables, nor does it explain whether the regression model assumptions were met. Thank you for your comment. We added the multicollinearity, and the models were fit in the Results section (page 5, lines 163-166)
response: It is recommended to supplement the model diagnostics, such as VIF values, residual analysis, etc., to improve the reliability of statistical inferences. We added the VIF < 5 in the Results section (page 5, lines 163-166)
Additionally, some conclusions do not closely correspond to the data analysis results. It is suggested to strengthen the logical connection between data analysis and conclusions to avoid excessive inferences.
response:
Thank you for highlighting. We have added corresponding data analysis results in the conclusion (line 299-305)
Overall, this article provides valuable insights into the mental health issues faced by frontline workers during public health emergencies. However, to improve the quality of the research, it is recommended that the author make systematic revisions to the research design, variable operationalization, statistical methods, and conclusion derivation. Future studies could also consider integrating qualitative interview methods to further explore the specific stress experiences and coping strategies of frontline workers, thereby providing a basis for developing more targeted mental health intervention measures.
response:
Thank you for the comments. We have refined the research design sentences for better clarity and defined the variables, accordingly, including the statistical methods and the conclusion section. (lines 77-141, 299-308)
This manuscript is a resubmission of an earlier submission. The following is a list of the peer review reports and author responses from that submission.
Round 1
Reviewer 1 Report
Comments and Suggestions for Authors
The research presents results from a non-representative sample survey of healthcare workers in a region of Malaysia during the early Covid-19 outbreak. The results are neither timely, comprehensive (e.g. lacking any comparators), nor generalizable (based on an online survey of dubious quality, e.g. low rate of valid responses). The literature is overly generic and dated, with minimal insights into the now-widespread evidence on how contemporary health system responses to the pandemic influenced health workforce surge capacity, stressors, and retention. Moreover, the analytical methods are very basic, describing more the statistical procedures (e.g. use of “enter” command) than any underlying conceptual framework to guide the analytics choices. It is unclear how this work would contribute to the global literature to help inform any form of healthcare policy, practice or research.
Reviewer 2 Report
Comments and Suggestions for Authors
While this paper is clearly of relevance to the healthcare services community, there are a few issues to be addressed/expounded on.
1. Ethical concerns - the constitution of the MHPSS team should be described in detail. Are they staff members or a separate research group and how are they aligned to the Selangor State Health Dept where the study was conducted? Could this have affected the participation of the target respondents in any way ?
2. How was anonymity of respondents assured?
3. Describe recruitment and consenting process in details.
4. Describe the sampling/selection process in details. Lack of representativeness of the sample would render biased results.
5. It would have been informative to see analysis of results on depression severity (DASS scale) by demographic data - i.e. age, sex, employment category, etc.
Reviewer 3 Report
Comments and Suggestions for Authors
Thank you for submitting your manuscript to this journal. Please find my comments below:
We are currently in 2025, approximately five years after your data gathering. The situation has changed significantly. How do you think your results and interpretation are still valid?
I think you are probably aware of several similar studies in the literature. Please describe how it is different from the others.
What were the variables that you adjusted the regression for?